# SHARP: Splatting High-fidelity And Relightable Photorealistic 3D Gaussian Head Avatars

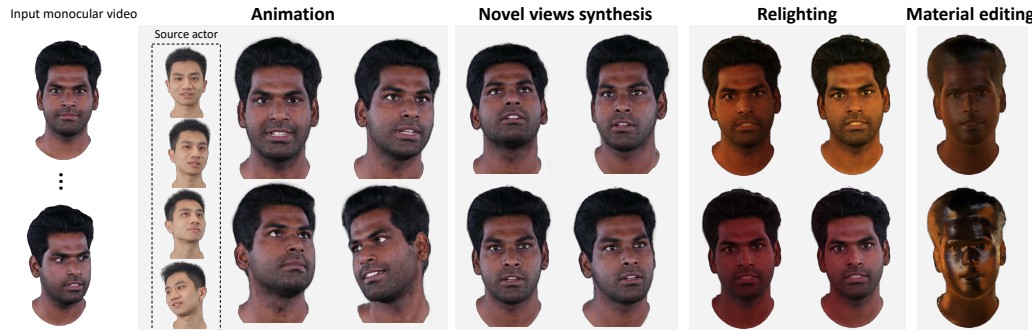

Figure 1: With monocular video input, SHARP reconstructs a high-fidelity, animatable 3D head avatar that enables realistic relighting effects and simple material editing.

## Abstract

Reconstructing animatable and high-fidelity 3D head avatars from monocular videos, especially with realistic relighting, is a valuable task. However, the limited information from single-view input, combined with the complex head poses and facial movements, makes this challenging. Previous methods achieve real-time performance by combining 3D Gaussian Splatting with a parametric head model, but the resulting head quality suffers from inaccurate face tracking and limited expressiveness of the deformation model. These methods also fail to produce realistic effects under novel lighting conditions. To address these issues, we propose SHARP, a method that reconstructs high-fidelity, relightable 3D head avatars using 3D Gaussian points. SHARP reduces tracking errors through end-to-end optimization and better captures individual facial deformations using learnable blendshapes and linear blend skinning. Additionally, it decomposes head appearance into several physical properties and incorporates physically-based shading to account for environmental lighting. Extensive experiments demonstrate that SHARP not only reconstructs superior-quality heads but also achieves realistic visual effects under varying lighting conditions.

## 1 Introduction

Creating a 3D head avatar is essential for film, gaming, immersive meetings, AR/VR, etc. In these applications, the avatar must meet several requirements: animatable, real-time, high-quality, and visually realistic. However, achieving a highly realistic and animatable head avatar from widely-used monocular video remains challenging.

Research in this area spans many years. Early efforts (Li et al., 2017; Paysan et al., 2009; Cao et al., 2013) develop parametric head models based on 3D Morphable Models (3DMM) theory (Blanz & Vetter, 1999). These methods allow registering 3D head scans to parametric models for 3D facial mesh reconstruction. With the rise of deep learning, methods (Tuan Tran et al., 2017; Chang et al., 2017; Daněček et al., 2022; Zielonka et al., 2022) use parametric model priors to simplify head

mesh reconstruction from videos, either through estimation or frame-wise optimization, *i.e.*, 3D face tracking. While these methods generalize well for expressions and pose variations, their fixed topology limits complex hair modeling and fine-grained appearance reconstruction. To address this issue, some researchers have turned to Neural Radiance Fields (NeRF)(Mildenhall et al., 2020) for modeling head avatars(Grassal et al., 2022; Qin et al., 2024b). These approaches enable complete geometry and appearance reconstruction, including hair, glasses, earrings, *etc.* However, they are limited by slow rendering and long training time. Recently, 3D Gaussian Splatting (3DGS)(Kerbl et al., 2023) has gained significant attention for its fast rendering speed. Some methods(Xiang et al., 2024; Shao et al., 2024) have extended 3DGS to head avatar reconstruction, significantly improving rendering speed compared to NeRF-based methods.

Although previous methods have made progress in animatability and real-time rendering, their reconstruction quality still suffers from two major factors: **1) Limited deformation flexibility** and **2) Inaccurate tracking**. Additionally, they are **unable to produce realistic relighting effects**. **First**, head reconstruction is dynamic, requiring a geometric model to deform from a compact canonical space to various states based on signals for different expressions and poses. Advanced methods (Xiang et al., 2024; Shao et al., 2024) model geometric deformations of Gaussian points by rigging them to universal parametric model mesh faces. However, parametric models may not accurately capture individuals' unique deformations, which restricts flexibility. **Second**, pseudo-2D key points are used to track expression and pose parameters before training. Insufficient accuracy in these key points and uncertainties in the 2D optimization process can lead to errors in tracked parameters, ultimately reducing reconstruction quality. Methods like Point-avatar(Zheng et al., 2023) optimize these parameters during training to minimize errors, which may create a mismatch with pre-tracked parameters, limiting generalization to new expressions and poses. Consequently, further optimization is often needed during testing. **Lastly**, under monocular and unknown lighting conditions, existing methods typically model appearance by fitting colors without accounting for external factors, which fails to simulate the true visual effects under varying lighting conditions.

To address these challenges, we propose SHARP, which utilizes 3D Gaussian points for high-quality head avatar reconstruction with realistic relighting from monocular video. Unlike previous rigging methods, we propose learnable blendshapes and learnable linear blend skinning, allowing the Gaussian points for flexible deformation from canonical space to pose space. Additionally, we utilize an encoder to extract accurate facial expression parameters from images and integrate the encoder into reconstruction training. This end-to-end optimization not only reduces the impact of tracking errors on reconstruction but also ensures the generalization of expression parameters estimation. To achieve realistic relighting, we model the head's appearance by using albedo, roughness, and Fresnel reflectance, shading images with a physically-based shading model. An albedo pseudo-prior is also employed to better decouple the albedo. Benefiting from these techniques, SHARP can reconstruct fine-grained and expressive avatars while accurately simulating realistic relighting effects.

In summary: **a)** We present SHARP, a method for monocular reconstruction of head avatars using 3D Gaussian points. SHARP leverages learnable blendshapes and learnable linear blend skinning for flexible and precise geometric deformations, with end-to-end optimization reducing tracking errors for high-quality reconstructions. **b)** We incorporate intrinsic priors to model head appearance under unknown lighting conditions. Combined with a physically-based shading model, we achieve realistic lighting effects across different environments. **c)** Experimental results demonstrate that SHARP outperforms existing methods in overall quality, enabling realistic relighting and simple material editing.

## 2 RELATED WORK

### 2.1 3D RADIANCE FIELDS

Image-based 3D reconstruction has become a vibrant research area due to its photorealistic visuals. NeRF(Mildenhall et al., 2020) introduced a novel method using MLPs to represent a 3D scene as a continuous density and color field, enabling differentiable image rendering through volume rendering. This approach has inspired numerous follow-up studies (Martin-Brualla et al., 2021; Yu et al., 2021; Barron et al., 2021; Wang et al., 2021). However, NeRF faces significant computational challenges due to extensive MLP queries. Instant-NGP(Müller et al., 2022) employs multi-resolution hash encoding to speed up inference. Additionally, some methods, propose hybrid 3D representa-

tions(Chan et al., 2022; Cao & Johnson, 2023; Fridovich-Keil et al., 2023) to improve efficiency. Recently, 3DGS introduces an explicit representation using Gaussian points, achieving real-time rendering with an efficient tile-based rasterizer. It rapidly gains attention, and researchers applying it to various fields(Wu et al., 2024a; Qin et al., 2024a; Zhang et al., 2024; Yu et al., 2024; Charatan et al., 2024; Huang et al., 2024) to exploit its rendering efficiency. Our work also builds upon 3DGS to achieve real-time rendering.

## 2.2 3D HEAD RECONSTRUCTION

3D head reconstruction broadly generally falls into two categories: geometric mesh reconstruction and novel view image synthesis. Traditional 3DMM(Blanz & Vetter, 1999) uses Principal Component Analysis (PCA) to create a parameterized facial model that represents appearance and geometric variations in a linear space. BFM(Paysan et al., 2009) improves on this by adding more scanned facial data, resulting in a richer model. FLAME(Li et al., 2017) introduces extra joints for the eyes, jaw, and neck, enabling more realistic facial motion. Deca(Feng et al., 2021) builds on FLAME by estimating parameters like shape and pose from a single image and capturing finer wrinkles through UV displacement maps. SMIRK(Retsinas et al., 2024) enhances tracking accuracy by using an image-to-image module to provide more precise supervision signals.

Recent advances in neural radiance fields combine 3DMM for view-consistent, photorealistic 3D head reconstruction. NeRFace(Gafni et al., 2021) extends NeRF to dynamic forms by incorporating expression and pose parameters as conditional inputs, enabling animatable head reconstruction. IMavatar(Zheng et al., 2022) models deformation fields for expression and pose motions, using iterative root-finding to locate the canonical surface intersection for each pixel. Point-avatar(Zheng et al., 2023) introduces a novel point-based representation with continuous deformation fields for more efficient animatable avatars. INSTA(Zielonka et al., 2023) speeds up training by using multi-resolution hashing for 3D head representation, deforming points based on the nearest mesh triangles. Recent works(Qian et al., 2024; Xiang et al., 2024) based on 3DGS achieve significant breakthroughs in rendering speed. 3D Gaussian Blendshapes(GBS)(Ma et al., 2024) learn Gaussian basis for blendshapes but struggle with pose variations. Our method enhances the reconstruction quality and provides realistic relighting effects, offering further advancements in these areas.

In addition to monocular methods, some researchers (Xu et al., 2024a; Giebenhain et al., 2024) explore multi-view video-based head reconstruction. However, these approaches require multiple synchronized cameras, making them more complex and less convenient than single-phone captures. Moreover, generative methods(Xu et al., 2024b; Kirschstein et al., 2024; An et al., 2023) can create 3D head avatars from a single image, providing another reconstruction approach.

## 2.3 NEURAL RELIGHTING

Implementing relighting in reconstructed 3D scenes is difficult. Some methods (Zhang et al., 2021b; Gao et al., 2020; Xu et al., 2023) use learning-based approaches to learn relightable appearances from images under varying lighting. In contrast, inverse rendering methods (Zhang et al., 2021c;a; Cai et al., 2022; Zhang et al., 2022) leverage reflection models like BRDF for more realistic relighting. Recent works(Gao et al., 2023; Jiang et al., 2024) integrate BRDF into 3DGS for real-time relighting and methods Wu et al. (2024b); Ye et al. (2024) introduce deferred shading for efficient relighting or specular rendering. Our approach also employs deferred shading for its effectiveness. Although some researchers combine physical reflection models with dynamic radiance fields to achieve relightable head avatars(Li et al., 2022; Yang et al., 2024; Saito et al., 2024), they require data under controlled lighting conditions. Reconstructing relightable 3D head avatars under monocular unknown lighting is still underexplored. Point-avatar models lighting but relies on trained shading networks, limiting its generalization. Our method enables relighting using new environment maps. While simplified physical rendering models can be inaccurate, many methods (Wu et al., 2024b; Jin et al., 2023; Li et al., 2024) add fitting-based rendering branches to improve results. We utilize physical rendering methods alone, achieving comparable effects without redundancy.

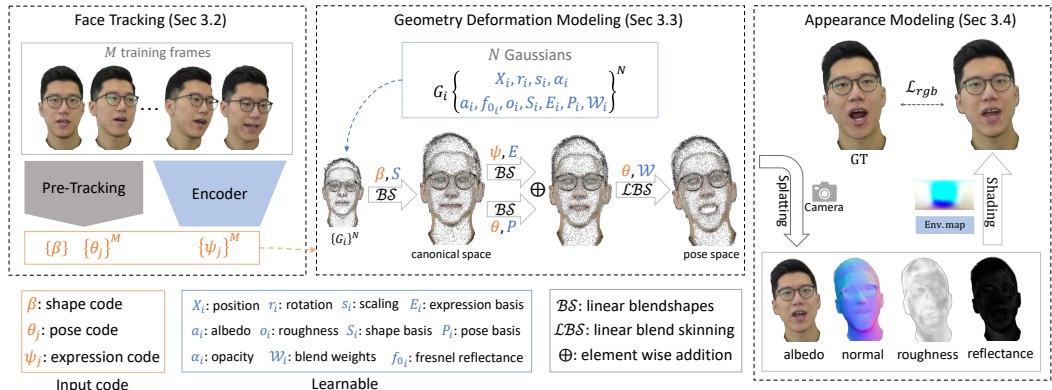

Figure 2: Given a monocular video with unknown lighting and $M$ frames, we first assume fixed camera parameters. We pre-track fixed shape parameter $\beta$ and pose parameters $\{\theta_j\}^M$ through iterative optimization. Expression parameters $\{\psi_j\}^M$ are inferred using an encoder which is optimized during training. With these parameters, we transform the Gaussian points into pose space using learnable linear blendshapes and linear blend skinning. We then render the Gaussian points to obtain albedo, roughness, reflectance, and normal maps. Finally, we compute pixel colors using physically-based shading with optimizable environment maps.

## 3 METHOD

As mentioned, previous methods for head reconstruction suffer from inaccurate tracking and deformation models with limited expressiveness. They also cannot achieve realistic relighting effects. To tackle these challenges, we enhance tracking accuracy through end-to-end optimization (Sec.3.2). We also introduce adaptive learning-based linear blendshapes and blend skinning for more flexible deformation of Gaussian points (Sec.3.3). Physically-based shading is employed to realistically model head appearance and achieve relighting (Sec.3.4). Finally, specific loss functions are utilized for training (Sec.3.5). The pipeline is illustrated in Fig.2.

### 3.1 PRELIMINARY

3D Gaussian Splatting (Kerbl et al., 2023) represents 3D scene with explicit Gaussian points, each point $G$ is defined by its position (center) $X$, rotation $r$, scaling $s$, opacity $\alpha$ and color $c$. During rendering, each Gaussian point affects nearby pixels anisotropically using a Gaussian function $\mathcal{G}$:

$$\mathcal{G}(x, \mu', \Sigma_{2D}) = e^{-\frac{1}{2}(x-\mu')^\top \Sigma_{2D}^{-1}(x-\mu')}, \tag{1}$$

where $\mu'$ is the projected mean of $X$ on the image plane. Given the viewing transformation $W$, the 2D covariance matrix $\Sigma_{2D}$ is derived from the 3D covariance matrix:

$$\Sigma_{2D} = JW\Sigma W^\top J^\top, \ \Sigma = RSS^\top R^\top. \tag{2}$$

$J$ is the Jacobian of the affine approximation of the projective transformation. To ensure the covariance matrix $\Sigma$ remains positive semi-definite during optimization, it is decomposed into a scaling matrix $S$ and a rotation matrix $R$, as Eq.2. The scaling matrix $S$ and rotation matrix $R$ are represented by a 3D vector $s$ and a quaternion $r$, respectively. The color $c$ is modeled by a third-order spherical harmonic coefficient for view-dependent effects. During splatting, the image space is divided into multiple $16 \times 16$ tiles and pixel colors are computed with alpha blending:

$$\mathcal{C}(x_p) = \sum_{i \in G_{x_p}} c_i \sigma_i \prod_{j=1}^{i-1}(1 - \sigma_j), \ \sigma_i = \mathcal{G}(x_p, \mu_i', \Sigma_{2D,i})\alpha_i, \tag{3}$$

where, $x_p$ represents the pixel position, and $G_{x_p}$ denotes the sorted Gaussian points associated with pixel $x_p$. Additionally, a gradient-based strategy is proposed to adjust the number of Gaussian points through densification and pruning.

### 3.2 FACE TRACKING

Current tracking methods estimate expression parameters with insufficient accuracy. Since these parameters control head expressions, inaccuracies can cause deformation errors, compromising reconstruction quality. To mitigate this issue while maintaining good generalization, we propose to use an encoder $\mathcal{E}$ to extract expression parameters from image $I$ and optimize it end-to-end during reconstruction. This enhances the encoder's inference accuracy and ensures better generalization:

$$\psi, \theta^{jaw} = \mathcal{E}(I), \tag{4}$$

where $\psi$ and $\theta^{jaw}$ represent the expression and jaw pose parameters, respectively. To prevent overfitting in jaw pose estimation, we introduce a regularization loss that constrains the distance between the inferred and the pre-tracked jaw poses $\hat{\theta}^{jaw}$:

$$\mathcal{L}_{jaw} = \left\| \hat{\theta}^{jaw} - \theta^{jaw} \right\|_2. \tag{5}$$

For simplicity, the pipeline (Fig.2) does not detail the jaw pose. Since no accurate method exists for full pose inference, other pose parameters in $\theta$ are pre-tracked. Furthermore, shape parameters $\beta$ are pre-tracked and are shared across all frames.

### 3.3 GEOMETRY DEFORMATION MODELING

Like most methods, we employ a deformation model to map points from canonical space to pose space based on expression and pose parameters. However, facial shapes, expressions, and pose deformations vary widely among individuals, making it difficult for parametric head models to accurately recover each person's unique shape and deformations. Simply rigging points on a parametric model limits expressive capacity. To flexibly model these distinct facial shapes and deformations, we propose adaptive learnable linear blendshapes and linear blend skinning for geometric deformation.

**Adaptively learnable linear blendshapes.** Similar to FLAME (Li et al., 2017), we use linear blendshapes to model geometric displacement. For each Gaussian point, we introduce three additional attributes: shape basis $S = \{S^1, ..., S^{|\beta|}\} \in \mathbb{R}^{N \times 3 \times |\beta|}$, expression basis $E = \{E^1, ..., E^{|\psi|}\} \in \mathbb{R}^{N \times 3 \times |\psi|}$ and pose basis $P = \{P^1, ..., P^{9K}\} \in \mathbb{R}^{N \times 3 \times 9K}$. These are learnable parameters that fit the individual head shape and deformations. First, we compute the shape offset to displace the points to the canonical space $X_c$ using shape blendshapes:

$$\mathcal{BS}(\beta, S) = \sum_{m=1}^{|\beta|} \beta^m S^m, \; X_c = X + \mathcal{BS}(\beta, S), \tag{6}$$

where $\mathcal{BS}$ denotes linear blendshapes and $\beta = \{\beta^1, ..., \beta^{|\beta|}\} \in \mathbb{R}^{|\beta|}$ is the shape parameter. Next, we compute expression and pose offsets in the same manner, using expression blendshapes and pose blendshapes to model facial expressions:

$$\mathcal{BS}(\psi, E) = \sum_{m=1}^{|\psi|} \psi^m E^m, \; \mathcal{BS}(\theta^*, P) = \sum_{m=1}^{9K} (\mathcal{R}(\theta^*)_m - \mathcal{R}(\theta^o)))P^m, \tag{7}$$

$$X_e = X_c + \mathcal{BS}(\psi, E) + \mathcal{BS}(\theta^*, P)), \tag{8}$$

where $\psi = \{\psi^1, ..., \psi^{|\psi|}\} \in \mathbb{R}^{|\psi|}$ is the expression parameter, and $\theta \in \mathbb{R}^{3(K+1)}$ is the pose parameter representing the axis-angle rotation of the points relative to the joints. $\theta^*$ excludes the global joint, with $K = 4$. $\mathcal{R}(\theta)$ is the flattened rotation matrix vector obtained by Rodrigues' formula, and $\theta^o$ represents the zero pose.

**Adaptively learnable linear blend skinning.** After applying linear displacement, we transform Gaussian points into pose space using Linear Blend Skinning (LBS). Each Gaussian point has a learnable blend weight $\mathcal{W} \in \mathbb{R}^{N \times K}$ to accommodate individual pose deformations. LBS rotates the points $X_e$ around each joints $\mathcal{J}(\beta)$ and linearly weighted by $\mathcal{W}$, defined as:

$$X_p = \mathcal{LBS}(X_e, \mathcal{J}(\beta), \mathcal{W}) = T_{lbs} X_e, \tag{9}$$

where $\mathcal{J}(\beta) \in \mathbb{R}^{K \times 3}$ represents the positions of the neck, jaw, and eyeball joints. To maintain geometric consistency, the rotation attributes of the Gaussians are also transformed by the weighted transformation matrix $T_{lbs}$: $R_p = T_{lbs} R$.

**Geometry initialization.** To facilitate easier learning, we leverage FLAME's geometric and deformation priors. We initialize the positions of the Gaussian points through linear interpolation on the FLAME mesh faces. The same method is applied to initialize the blendshapes basis and blendweights. Other geometric attributes, like rotation and scale, are initialized similarly to 3DGS.

### 3.4 Appearance Modeling

3DGS uses spherical harmonics to model the view-dependent appearance of each point, but it cannot simulate visual effects under new lighting conditions. To overcome this, we introduce a novel appearance modeling approach that decomposes the appearance into three properties: albedo $a$, roughness $o$, and Fresnel base reflectance $f_0$. We then utilize a BRDF model(Burley & Studios, 2012) for physically-based shading of the image. To enhance efficiency, we apply the SplitSum approximation technique (Karis & Games, 2013) to precompute the environment map.

**Shading.** First, we render the albedo map $\mathbf{A}$, roughness map $\mathbf{O}$, reflectance map $\mathbf{F_0}$, and normal map $\mathbf{N}$ using rasterizer. The specular and diffuse maps are then calculated as follows:

$$I_{specualr} = I_{env}(\mathbf{R}, \mathbf{O}) \cdot (ks \cdot I_{BRDF}(\mathbf{O}, \mathbf{N} \cdot \mathbf{V})[0] + I_{BRDF}(\mathbf{O}, \mathbf{N} \cdot \mathbf{V})[1]), \tag{10}$$

$$I_{diffuse} = \mathbf{A} \cdot I_{irr}(\mathbf{N}), \tag{11}$$

where $\mathbf{V}$ is the view direction map derived from the camera parameters and $\mathbf{R}$ is the reflection direction map, computed as $\mathbf{R} = 2(\mathbf{N} \cdot \mathbf{V})\mathbf{N} - \mathbf{V}$. $I_{BRDF}$ is a precomputed map of the simplified BRDF integral. We use an approximate Fresnel equation $\tilde{\mathcal{F}}$ to compute the specular reflectance $ks$:

$$ks = \tilde{\mathcal{F}}(\mathbf{N} \cdot \mathbf{V}, \mathbf{O}, \mathbf{F_0}) = \mathbf{F_0} + (max(1 - \mathbf{O}, \mathbf{F_0}) - \mathbf{F_0}) \cdot 2^{(-5.55473(\mathbf{N} \cdot \mathbf{V}) - 6.698316) \cdot (\mathbf{N} \cdot \mathbf{V})}. \tag{12}$$

The final shaded image is computed as: $I_{shading} = I_{diffuse} + I_{specular}$. During training, we optimize two cube maps: the environment irradiance map $I_{irr}$ and the prefiltered environment map $I_{env}$. $I_{env}(\mathbf{R}, \mathbf{O})$ provides radiance values based on the reflection directions and roughness, while $I_{irr}(\mathbf{N})$ provides irradiance values based on the normal directions.

**Normal estimation.** Smooth and accurate normals are essential for physical rendering, as rough normals can cause artifacts during relighting. Following Jiang et al. (2024), we use the shortest axis of each Gaussian point as its normal $n$. To ensure the correct direction and geometric consistency, we supervise the rendered normal map $\mathbf{N}$ with the normal map $\hat{\mathbf{N}}$ obtained from depth derivatives:

$$\mathcal{L}_{normal} = \left\| \mathbf{1} - \mathbf{N} \cdot \hat{\mathbf{N}} \right\|_1. \tag{13}$$

**Intrinsic prior**. Disentangling material properties under constant unknown lighting is challenging due to inherent uncertainties. When reconstructing heads under non-uniform lighting, local lighting effects can be erroneously coupled into the albedo, resulting in unrealistic relighting. To address this, we use the existing model Chen et al. (2024) to extract pseudo-ground-truth albedos $\mathbf{A}^{gt}$, supervising the rendered albedos for a more realistic appearance, as Eq.14. We also constrain the roughness and base reflectance within predefined ranges: $o \in [\tau_{min}^o, \tau_{max}^o]$, $f_0 \in [\tau_{min}^{f_0}, \tau_{max}^{f_0}]$.

$$\mathcal{L}_{albedo} = \left\| \mathbf{A} - \mathbf{A}^{gt} \right\|_1. \tag{14}$$

### 3.5 Optimization

During optimization, we retain the point densification and pruning strategy from 3DGS, with additional attributes inherited similarly. In addition to the previously mentioned losses, we use the Mean Absolute Error (MAE) and a D-SSIM to calculate the error between the rendered image and ground truth, as Eq.16. We also apply Total Variation (TV) loss $\mathcal{L}_{tv}$ to the rendered roughness map $\mathbf{O}$ to ensure smoothness. The total loss function is given in Eq.15. The weights for each loss component are set as follows: $\lambda_{jaw} = 0.1$, $\lambda_1 = 0.8$, $\lambda_{\mathcal{W}} = 0.1$, $\lambda_{normal} = 10^{-5}$, $\lambda_{albedo} = 0.25$, $\lambda_{tv} = 0.02$.

$$\mathcal{L}_{total} = \mathcal{L}_{rgb} + \lambda_{jaw}\mathcal{L}_{jaw} + \lambda_{normal}\mathcal{L}_{normal} + \lambda_{albedo}\mathcal{L}_{albedo} + \lambda_{tv}\mathcal{L}_{tv}(\mathbf{O}), \tag{15}$$

$$\text{where} \quad \mathcal{L}_{rgb} = \lambda_1 \left\| I_{shading} - I_{gt} \right\|_1 + (1 - \lambda_1)\mathcal{L}_{\text{D-SSIM}}(I_{shading}, I_{gt}). \tag{16}$$

Table 1: Average quantitative results on the INSTA, HDTF, and self-captured datasets. Our method outperforms others in PSNR, MAE* (MAE $\times 10^2$), SSIM, and LPIPS metrics.

| Method | INSTA dataset | | | | HDTF dataset | | | | self-captured dataset | | | |
|---|---|---|---|---|---|---|---|---|---|---|---|---|
| | PSNR↑ | MAE*↓ | SSIM↑ | LPIPS↓ | PSNR↑ | MAE*↓ | SSIM↑ | LPIPS↓ | PSNR↑ | MAE*↓ | SSIM↑ | LPIPS↓ |
| INSTA | 27.85 | 1.309 | 0.9110 | 0.1047 | 25.03 | 2.333 | 0.8475 | 0.1614 | 25.91 | 1.910 | 0.8333 | 0.1833 |
| Point-avatar | 26.84 | 1.549 | 0.8970 | 0.0926 | 25.14 | 2.236 | 0.8385 | 0.1278 | 25.83 | 1.692 | 0.8556 | 0.1241 |
| Splatting-avatar | 28.71 | 1.200 | 0.9271 | 0.0862 | 26.66 | 2.01 | 0.8611 | 0.1351 | 26.47 | 1.711 | 0.8588 | 0.1550 |
| Flash-avatar | 29.13 | 1.133 | 0.9255 | 0.0719 | 27.58 | 1.751 | 0.8664 | 0.1095 | 27.46 | 1.632 | 0.8348 | 0.1456 |
| GBS | 29.64 | 1.020 | 0.9394 | 0.0823 | 27.81 | 1.601 | 0.8915 | 0.1297 | 28.59 | 1.331 | 0.8891 | 0.1560 |
| SHARP (Ours) | **30.36** | **0.845** | **0.9482** | **0.0569** | **28.55** | **1.373** | **0.9089** | **0.0825** | **28.97** | **1.123** | **0.9054** | **0.1059** |

Table 2: Ablation quantitative results on the INSTA dataset. **Bold** marks the best results, and underline marks the second best results.

| | full (ours) | rigged to FLAME | w/o encoder | w/o learnable | w/o PBS |
|---|---|---|---|---|---|
| PSNR↑ | **30.36** | 29.79 | 29.70 | 29.83 | 30.34 |
| MAE*↓ | **0.845** | 0.937 | 0.933 | 0.923 | 0.850 |
| SSIM↑ | **0.9482** | 0.9431 | 0.9438 | 0.9440 | 0.9480 |
| LPIPS↓ | 0.0569 | 0.0695 | 0.0667 | 0.0684 | **0.0563** |

## 4 EXPERIMENT

### 4.1 EXPERIMENTAL SETUP

**Implementation details.** We build our model using PyTorch (Paszke et al., 2019) and train it with the Adam optimizer (Kingma, 2014) on a single NVIDIA 3090 GPU. Each monocular head video is trained for 15 epochs. All videos are cropped and resized to a resolution of $512 \times 512$. We use RVM (Lin et al., 2022) to extract the foreground, setting the background to black. Moreover, we follow Zheng et al. (2022) to pre-track FLAME parameters for the videos. For our encoder $\mathcal{E}$, we utilize the pre-trained weight from SMIRK (Retsinas et al., 2024).

**Dataset.** We evaluate different methods on 10 subjects from the INSTA dataset (Zielonka et al., 2023), which provides pre-cropped and segmented images. Following INSTA, we use the last 350 frames of each video as the test set for self-reenactment evaluation. For a more robust assessment, we include 8 subjects from the HDTF dataset (Zhang et al., 2021d), which is collected from the internet. We also include 5 self-captured subjects using a mobile phone. For these two datasets, the last 500 frames are used as the test set. All methods adopt the same cropped and segmented process.

**Baseline and metrics.** We compare our method against several SOTA methods: Point-avatar(Zheng et al., 2023), INSTA(Zielonka et al., 2023), Splatting-avatar(Shao et al., 2024), Flash-avatar(Xiang et al., 2024), and 3D Gaussian Blendshapes (GBS)(Ma et al., 2024). For each method, we use the tracking approach described in their papers. Note that we disable the post-training optimization of test images' parameters in Point-avatar to ensure fairness. We use PSNR, MAE* (MAE $\times 10^2$), SSIM, and LPIPS (Zhang et al., 2018) to evaluate the image quality.

### 4.2 EVALUATION

**Quantitative results.** We evaluate all methods for self-reenactment, as shown in Tab.1. Our method outperforms others across all three metrics, especially in LPIPS. This highlights that our method reconstructs more detailed and high-quality animatable avatars, with the improved LPIPS score suggesting sharper images.

**Qualitative results.** The visual comparison of our method with baseline methods on self-reenactment is shown in Fig.3. INSTA and Splatting-avatar often struggle with challenging poses, resulting in significant artifacts. Point-avatar maintains decent rendering in such poses but suffers from point artifacts and lacks detail in the mouth. Flash-avatar shows improvements but still loses some fine textures and has expression inaccuracies. GBS achieves relatively accurate facial expressions in normal poses but introduces blurring around edges, like the ears, hair, and neck. In contrast, our method accurately restores fine textures, such as hair and eye luster, while preserving precise

Figure 3: Qualitative comparison results on self-reenactment. Compared to others, ours captures finer texture details and renders sharper images. Ours also achieves more accurate expression deformations and reconstructs better geometric details.

geometric details like ears and teeth. Ours handles wrinkles and blinking more effectively due to the flexible deformation model and accurate tracking.

We also present cross-reenactment visual comparisons. As shown in Fig.4, our method better retains the source actor's expressions and preserves original head details, even in challenging poses and expressions, while other methods exhibit blurring and artifacts. It's worth noting that Flash-avatar and GBS treat head poses as camera poses, which may cause minor scale discrepancies, resulting in variations in the size and positioning of rendered avatars.

## 4.3 ABLATION STUDIES

The quantitative results of the ablation study on self-reenactment are summarized in Tab.2, with qualitative results in Fig.5 and Fig.6, validating the effectiveness of each component.

**Rigged to FLAME.** We replace SHARP's deformation model with the method from Qian et al. (2024), which rigs Gaussian points to the FLAME mesh. The results in Tab.2 and Fig.5 demonstrate that our model improves on metrics and achieves more accurate texture and tooth details.

**Without learnable.** We set the blendshapes basis and blendweights as non-learnable to assess the importance of adapting to individual deformations. This leads to decreased performance on metrics and reduced geometry and texture quality.

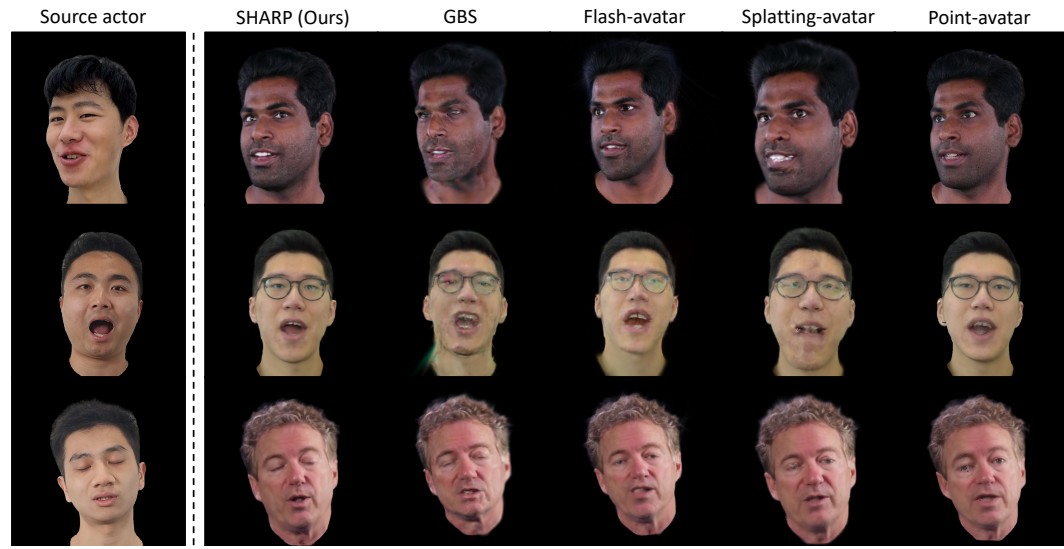

Figure 4: Visual comparison on cross-reenactment. SHARP accurately simulates actors' poses and expressions, preserving textures and geometric details, while others exhibit artifacts and blurring.

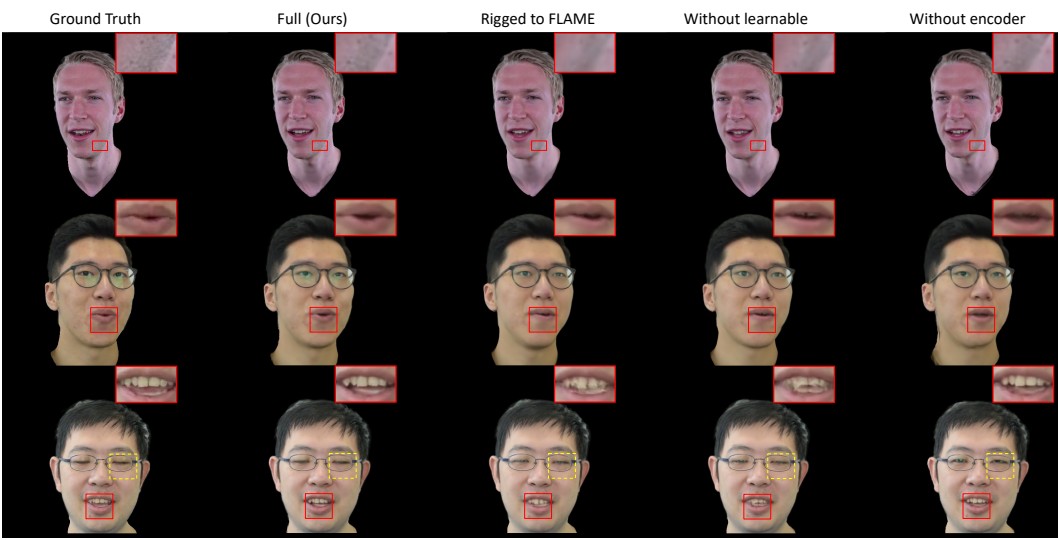

Figure 5: Qualitative results of the ablation study. Our full method renders better texture and geometry details and captures more accurate facial expressions, including mouth shapes and blinking.

**Without encoder.** To verify the end-to-end trained encoder's effectiveness in extracting expression parameters, we use pre-tracked parameters instead. Results indicate our method better restores facial expressions, including mouth shapes and blinking, and improves performance metrics.

**Without PBS.** This means using the standard 3DGS appearance model instead of our shading model. While the fitting-based method of 3DGS performs well due to more learnable parameters and flexibility, our method achieves comparable results while enabling realistic relighting.

**Without $\mathcal{L}_{normal}$.** As shown in Fig.6, removing normal consistency loss results in chaotic normal maps, causing blocky artifacts during relighting.

**Without $\mathcal{L}_{albedo}$.** Without the albedo prior loss, appearance attributes become entangled, causing incorrect coupling of local highlights with albedo. This results in unrealistic relighting effects, with highlights appearing in areas without actual lighting, as shown in Fig.6.

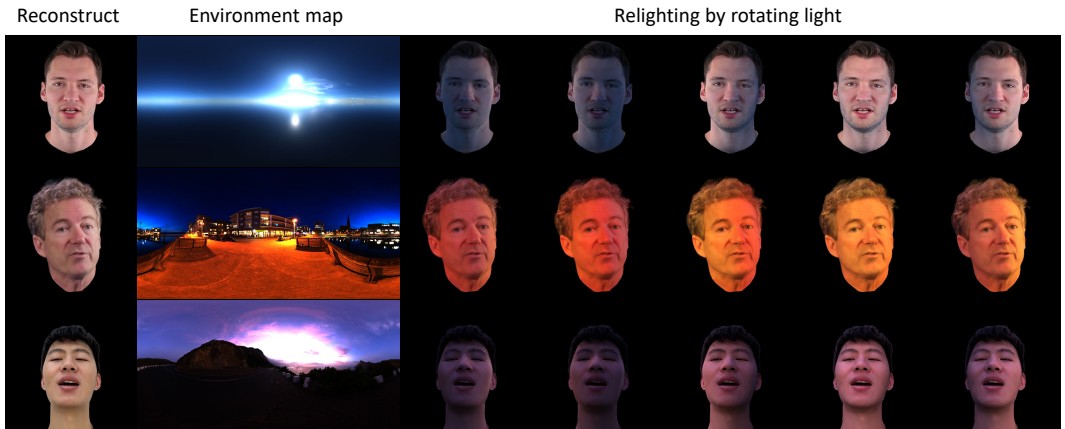

Figure 6: Ablation study for albedo and normal losses. Without $\mathcal{L}_{albedo}$, entangled attributes yield unrealistic relighting. Without $\mathcal{L}_{normal}$, chaotic normal maps cause artifacts when relighting.

Figure 7: Relighting visual results. For each environment map, we rotate the lighting to illuminate the head from different directions.

## 4.4 APPLICATION

**Relighting.** We show the relighting results of the head illuminated by rotating environment maps in Fig.7. For each map, we extract the corresponding irradiance and prefiltered maps, applying them in the shading process (Sec.3.4). Our method effectively simulates realistic visual effects.

**Material editing and novel view synthesis.** We present these results in appendices.

## 5 CONCLUSION

In this paper, we introduce SHARP, a novel method for high-fidelity, relightable 3D head avatar reconstruction from monocular video. To address errors incorporated from inaccurate facial expression tracking, we train an encoder in an end-to-end manner to extract more precise parameters. We model individual-specific deformations using learnable blendshapes and linear blend skinning for flexible Gaussian point deformation. By employing physically-based shading for appearance modeling, our method enables realistic relighting. Experimental results show that SHARP achieves state-of-the-art quality and realistic relighting effects.

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

## A  VIDEO DEMO

We strongly encourage readers to watch the video provided in the supplementary materials. It showcases the self-reenactment animation of avatars reconstructed by SHARP and includes novel view renderings. The video also illustrates the visual results of relighting the avatars under various rotating environment maps and the ability to perform simple material editing to enhance specular reflections. Additionally, we provide visual comparisons of SHARP with two advanced methods, GBS(Ma et al., 2024) and Flash-avatar(Xiang et al., 2024), in self-reenactment, cross-reenactment, and novel view synthesis. Overall, the video highlights our method's capability to create fine-grained avatars with excellent expressiveness and realistic lighting effects in diverse environments.

## B  MORE IMPLEMENTATION DETAILS

### B.1  TRAINING DETAILS

In the first 1500 iterations, we take the albedo map as the rendered image to learn the head's albedo properties initially. Afterward, we switch to shaded image to learn other attributes. While we generally follow 3DGS hyperparameters, we make some adjustments. During training, point densification starts at iteration 1000 and ends at 500 iterations before training completes, with a densification interval of 500 iterations. The gradient threshold is increased to $3 \times 10^{-4}$ to avoid excessive point growth. The learning rates for the Gaussian point positions, appearance attributes, and environment map gradually decrease as training progresses, while the encoder learning rate is set to $5 \times 10^{-5}$. Training a video with 2500 frames takes about one hour.

When using albedo prior to supervision, we apply it every 3 frames due to the time-consuming process of extracting pseudo-ground-truth albedo during preprocessing. Additionally, since the lighting in the INSTA and self-captured datasets is relatively uniform, we only apply albedo prior supervision during training on the HDTF dataset. Furthermore, for subjects in the HDTF dataset, we set a higher upper bound for reflectance ($\tau_{max}^{f_0}$) to account for the specific lighting conditions.

### B.2  MODEL DETAILS

The shape and expression basis in FLAME are derived through PCA, with higher dimensions having a small effect on deformation. To avoid unnecessary computations, we use only the first 100 shape parameters and 50 expression parameters, i.e., $|\beta| = 100$ and $|\psi| = 50$. Since FLAME lacks an interior mesh for the mouth, we follow Qian et al. (2024) by adding a mesh for the teeth, where the upper and lower teeth move according to the neck and jaw joints, respectively. Additionally, we add extra mesh behind the teeth to provide a reasonable initialization for the rest of the mouth interior.

During shading, normal and reflection vectors sample lighting from the irradiance and pre-filtered environment maps. Since both maps must be backpropagated and mipmaps reconstructed in each training iteration, the computation increases with resolution. To maintain efficient training, we set the irradiance map $I_{irr}$ resolution to $16 \times 16$ and the pre-filtered environment map $I_{env}$ to $32 \times 32$ with 3 mipmap levels.

### B.3  BRDF REFLECTION MODEL.

For physical-based shading, we use the Disney model(Burley & Studios, 2012) to describe light interactions with geometry and materials, a method commonly employed in real-time rendering. This model breaks reflection into two components: Lambertian diffuse reflection and specular reflection:

$$L_o(X, \omega_o) = L_d + L_s = \int_\Omega \frac{a}{\pi} L_i(X, \omega_i) n \cdot \omega_i d\omega_i + \int_\Omega \frac{\mathcal{D}\mathcal{F}\mathcal{H}}{4(n \cdot \omega_o)(n \cdot \omega_i)} L_i(X, \omega_i) n \cdot \omega_i d\omega_i, \quad (17)$$

where $L_i$ and $L_o$ denote the radiance for the incoming direction $\omega_i$ and outgoing direction $\omega_o$, respectively with $n$ as the normal. The Lambertian term models diffuse reflection, independent of viewing direction, allowing us to precompute and store this part in an irradiance map. The specular reflection term models appearance based on viewing angle, with $\mathcal{D}$, $\mathcal{F}$, and $\mathcal{H}$ representing the

Table 3: Complete quantitative results of self-reenactment for each subject on the INSTA dataset. SHARP achieves better performance metrics in most cases. **Bold** marks the best, and underline marks the second.

| | | Bala | biden | justin | malte_1 | marcel | nf_01 | nf_03 | obama | person0004 | wojtek_1 |
|---|---|---|---|---|---|---|---|---|---|---|---|
| | | | | | | | INSTA dataset | | | | |
| PSNR↑ | INSTA | 29.53 | 29.92 | 31.66 | 27.44 | 22.99 | 26.45 | 28.31 | 31.21 | 25.44 | 31.36 |
| | Point-avatar | 27.88 | 27.64 | 30.40 | 24.98 | 24.66 | 25.25 | 26.60 | 28.83 | 23.29 | 28.82 |
| | Splatting-avatar | 32.14 | 30.42 | 30.93 | 27.66 | 24.34 | 27.08 | 27.85 | 30.64 | 26.49 | 29.54 |
| | Flash-avatar | 30.27 | 31.25 | 32.16 | 27.45 | 24.85 | 28.02 | 28.28 | 31.46 | 25.49 | 32.03 |
| | GBS | 32.47 | **32.23** | 33.10 | 28.23 | 26.11 | 27.59 | 28.12 | 31.35 | 25.16 | **32.05** |
| | SHARP (Ours) | **33.10** | 31.70 | **33.29** | **29.28** | **26.58** | **28.95** | **29.68** | **33.24** | **26.54** | 31.26 |
| MAE*↓ | INSTA | 1.154 | 0.849 | 0.642 | 1.160 | 2.996 | 1.705 | 1.381 | 0.775 | 1.594 | 0.834 |
| | Point-avatar | 1.386 | 1.203 | 0.869 | 1.596 | 2.662 | 1.800 | 1.583 | 1.103 | 2.083 | 1.042 |
| | Splatting-avatar | 0.854 | 0.838 | 0.783 | 1.135 | 2.309 | 1.533 | 1.340 | 0.917 | 1.376 | 0.910 |
| | Flash-avatar | 1.175 | 0.670 | 0.610 | 1.058 | 2.133 | 1.326 | 1.249 | 0.819 | 1.589 | 0.700 |
| | GBS | 0.747 | 0.583 | 0.520 | 1.010 | 1.608 | 1.311 | 1.162 | 0.802 | 1.803 | **0.655** |
| | SHARP (Ours) | **0.657** | 0.616 | **0.498** | **0.902** | **1.293** | **1.133** | **1.031** | **0.580** | **1.070** | 0.668 |
| SSIM↑ | INSTA | 0.8896 | 0.9460 | 0.9591 | 0.9159 | 0.8736 | 0.8937 | 0.8676 | 0.9484 | 0.8478 | 0.9452 |
| | Point-avatar | 0.8658 | 0.9116 | 0.9373 | 0.8853 | 0.9063 | 0.8919 | 0.8807 | 0.9145 | 0.8576 | 0.9192 |
| | Splatting-avatar | 0.9272 | 0.9466 | 0.9482 | 0.9243 | 0.9041 | 0.9202 | 0.9113 | 0.9411 | 0.9075 | 0.9400 |
| | Flash-avatar | 0.8494 | 0.9614 | 0.9611 | 0.9326 | 0.9086 | 0.9270 | 0.9155 | 0.9493 | 0.8996 | 0.9509 |
| | GBS | 0.9390 | **0.9658** | **0.9690** | 0.9374 | 0.9217 | 0.9365 | 0.9271 | 0.9476 | 0.8910 | **0.9593** |
| | SHARP (Ours) | **0.9473** | 0.9635 | 0.9687 | **0.9429** | **0.9352** | **0.9398** | **0.9334** | **0.9647** | **0.9278** | 0.9590 |
| LPIPS↓ | INSTA | 0.0992 | 0.0541 | 0.0521 | 0.0731 | 0.1351 | 0.1262 | 0.1286 | 0.0446 | 0.1453 | 0.0540 |
| | Point-avatar | 0.0829 | 0.0637 | 0.0588 | 0.0758 | 0.1247 | 0.1257 | 0.1143 | 0.0589 | 0.1637 | 0.0576 |
| | Splatting-avatar | 0.0865 | 0.0564 | 0.0651 | 0.0749 | 0.1326 | 0.1107 | 0.0966 | 0.0545 | 0.1246 | 0.0602 |
| | Flash-avatar | 0.1535 | **0.0299** | 0.0378 | 0.0477 | 0.1069 | 0.0868 | 0.0760 | 0.0376 | 0.1035 | 0.0392 |
| | GBS | 0.0862 | 0.0433 | 0.0481 | 0.0737 | 0.1219 | 0.1076 | 0.0861 | 0.0564 | 0.1417 | 0.0582 |
| | SHARP (Ours) | **0.0451** | 0.0306 | **0.0367** | **0.0476** | **0.0992** | **0.0868** | **0.0649** | **0.0279** | **0.0940** | **0.0358** |

Table 4: Complete quantitative results of self-reenactment for each subject on the HDTF dataset. SHARP achieves better performance metrics in most cases.

| | | elijah | haaland | katie | marcia | randpaul | schako | tom | veronica | ckj | ft | lyf | zdb | zzy |
|---|---|---|---|---|---|---|---|---|---|---|---|---|---|---|
| | | | | | HDTF dataset | | | | | | | self-captured dataset | | |
| PSNR↑ | INSTA | 25.00 | 24.94 | 21.36 | 24.61 | 23.50 | 26.45 | 29.16 | 26.45 | 25.88 | 25.37 | 29.33 | 24.86 | 24.086 |
| | Point-avatar | 24.05 | 25.56 | 22.51 | 23.76 | 26.28 | 25.44 | 27.01 | 26.51 | 25.35 | 27.32 | 28.09 | 23.56 | 24.85 |
| | Splatting-avatar | 26.08 | 26.31 | 22.23 | 25.80 | 29.25 | 25.51 | 30.98 | 27.14 | 25.05 | 28.20 | 29.54 | 25.34 | 24.22 |
| | Flash-avatar | 26.29 | 26.46 | 23.39 | 26.67 | 29.05 | **28.28** | 31.56 | 28.95 | 26.37 | 27.26 | 30.59 | **28.01** | 25.09 |
| | GBS | 26.76 | 28.29 | 22.74 | 26.59 | 29.20 | 27.88 | 31.54 | 29.48 | 28.15 | 29.50 | **31.64** | 27.48 | 26.17 |
| | SHARP (Ours) | **28.24** | **28.91** | **24.92** | **27.23** | **29.70** | 27.95 | **31.75** | **29.71** | **29.40** | **30.19** | 31.40 | 27.00 | **26.84** |
| MAE*↓ | INSTA | 1.835 | 2.161 | 4.179 | 2.191 | 2.602 | 1.936 | 1.272 | 2.487 | 1.877 | 1.637 | 1.377 | 1.841 | 2.807 |
| | Point-avatar | 2.058 | 2.177 | 3.493 | 2.423 | 1.746 | 2.092 | 1.683 | 2.212 | 1.852 | 1.312 | 1.204 | 1.903 | 2.210 |
| | Splatting-avatar | 1.652 | 1.915 | 3.841 | 2.026 | 1.260 | 2.200 | 0.988 | 2.183 | 2.093 | 1.296 | 1.110 | 1.565 | 2.489 |
| | Flash-avatar | 1.602 | 2.052 | 2.922 | 1.755 | 1.312 | 1.519 | 0.980 | 1.865 | 1.909 | 1.364 | 1.079 | 1.251 | 2.557 |
| | GBS | 1.406 | 1.403 | 3.216 | 1.659 | 1.234 | 1.452 | 0.901 | 1.535 | 1.379 | 1.022 | 0.950 | 1.285 | 2.018 |
| | SHARP (Ours) | **1.108** | **1.319** | **2.283** | **1.483** | **1.079** | **1.384** | **0.847** | **1.477** | **1.142** | **0.896** | **0.792** | **1.117** | **1.666** |
| SSIM↑ | INSTA | 0.8808 | 0.8337 | 0.7474 | 0.8290 | 0.8528 | 0.8586 | 0.9143 | 0.7700 | 0.8218 | 0.8659 | 0.8722 | 0.8634 | 0.7431 |
| | Point-avatar | 0.8631 | 0.8275 | 0.7771 | 0.8160 | 0.8694 | 0.8578 | 0.8634 | 0.8339 | 0.8460 | 0.8763 | 0.8867 | 0.8573 | 0.8117 |
| | Splatting-avatar | 0.8952 | 0.8562 | 0.7562 | 0.8477 | 0.9094 | 0.8586 | 0.9321 | 0.8337 | 0.8279 | 0.8775 | 0.9038 | 0.8817 | 0.8031 |
| | Flash-avatar | 0.8898 | 0.8146 | 0.8133 | 0.8636 | 0.9040 | 0.8982 | 0.9305 | 0.8170 | 0.7774 | 0.8659 | 0.8967 | 0.8850 | 0.7491 |
| | GBS | 0.9113 | 0.8924 | 0.8068 | 0.8783 | 0.9110 | 0.9091 | 0.9404 | 0.8826 | 0.8799 | 0.9098 | 0.9188 | 0.9029 | 0.8339 |
| | SHARP (Ours) | **0.9335** | **0.9036** | **0.8597** | **0.8961** | **0.9254** | **0.9135** | **0.9446** | **0.8951** | **0.9019** | **0.9232** | **0.9283** | **0.9142** | **0.8596** |
| LPIPS↓ | INSTA | 0.1005 | 0.1698 | 0.2222 | 0.1586 | 0.1417 | 0.1390 | 0.0729 | 0.2415 | 0.1897 | 0.1583 | 0.1523 | 0.1678 | 0.2483 |
| | Point-avatar | 0.0886 | 0.1360 | 0.1683 | 0.1200 | 0.1147 | 0.1283 | 0.0981 | 0.1686 | 0.1255 | 0.0942 | 0.1024 | 0.1364 | 0.1623 |
| | Splatting-avatar | 0.0902 | 0.1476 | 0.1982 | 0.1385 | 0.1033 | 0.1455 | 0.0664 | 0.1907 | 0.1773 | 0.1271 | 0.1194 | 0.1539 | 0.1972 |
| | Flash-avatar | 0.0759 | 0.1595 | 0.1387 | 0.0881 | 0.0829 | 0.1011 | 0.0609 | 0.1688 | 0.2346 | 0.0736 | **0.0901** | **0.109** | 0.2208 |
| | GBS | 0.0875 | 0.1515 | 0.1899 | 0.1289 | 0.1113 | 0.1160 | 0.0679 | 0.1850 | 0.1696 | 0.1198 | 0.1305 | 0.1599 | 0.2004 |
| | SHARP (Ours) | **0.0504** | **0.0929** | **0.1208** | **0.0723** | **0.0683** | **0.0846** | **0.0485** | **0.12228** | **0.1063** | **0.0662** | 0.0939 | 0.1153 | **0.1478** |

normal distribution, Fresnel equation, and geometric function. We use the SplitSum approximation to simplify the BRDF integral into two parts:

$$L_s \approx \left(\frac{1}{\mathrm{Z}}\sum_{z=1}^{\mathrm{Z}} L_i(\omega_z)\right)\left(\frac{1}{\mathrm{Z}}\sum_{z=1}^{\mathrm{Z}} \frac{\mathcal{DFH} \cdot n \cdot \omega_z}{4(n \cdot \omega_o)(n \cdot \omega_z)pdf(\omega_z,\omega_o)}\right) = I_{env} \cdot I_{BRDF}. \quad (18)$$

Here, $pdf(\omega_m,\omega_o)$ is the probability density function related to $\mathcal{D}$. Both components are precomputed and stored: $I_{env}$ as a multi-resolution mipmap for different roughness levels and $I_{BRDF}$, as a lookup table (LUT) based on roughness and $n \cdot \omega_o$.

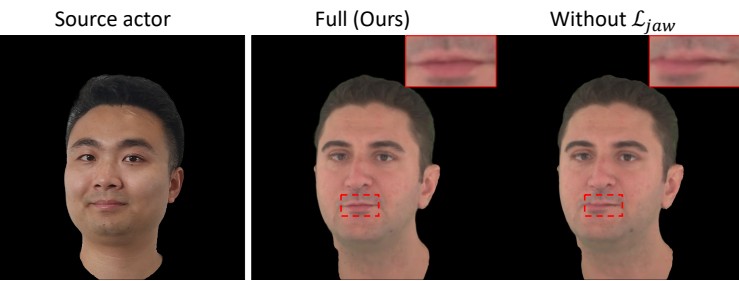

Figure 8: Ablation result on $\mathcal{L}_{jaw}$. Without the jaw pose regularization loss, the avatar exhibits mouth distortion during cross-reenactment.

## C  FURTHER EXPERIMENTS

### C.1  COMPLETE QUANTITATIVE RESULTS

We present the complete quantitative results of self-reenactment for each subject on the INSTA, HDTF, and self-captured datasets in Tab.3 and Tab.4. As shown, SHARP achieves superior performance for most subjects, demonstrating the robustness of our method.

### C.2  ABLATION ON $\mathcal{L}_{jaw}$

Without the jaw pose regularization loss, $\mathcal{L}_{jaw}$, the trained encoder may extract jaw poses that deviate from the normal distribution. This can lead to incorrect mouth motion during cross-reenactment. As shown in Fig.8, removing $\mathcal{L}_{jaw}$ results in mouth distortion, while including this loss effectively prevents the issue.

### C.3  RENDERING SPEED

Despite the additional computational load introduced by the deformation and appearance models, our method still achieves real-time rendering speeds. To provide a reference, we test the rendering speed on a subject from the INSTA dataset using a single NVIDIA 3090 GPU. This trained avatar contains 84,382 Gaussian points. We set the rendering resolution to $512 \times 512$ and render 500 images to calculate the average speed. SHARP reaches a **real-time rendering speed** of approximately **154 FPS** for this subject, with the encoder extracting parameters at about 179 FPS. Similarly, when relighting with a new environment map, we measured a rendering speed of approximately 154 FPS under the same setup, ensuring real-time performance.

## D  ADDITIONAL APPLICATIONS

### D.1  RELIGHTING

We present the relighting results of various avatars under rotating environment maps in Fig.7 of the main paper. Here, we provide additional details on the relighting implementation.

For convenience during relighting, we use off-the-shelf tools to precompute the irradiance map and pre-filtered environment map from the environment map. Specifically, we use CmftStudio, a tool commonly used in real-time rendering pipelines to process HDR images for image-based lighting. With CmftStudio, we extract the original environment map with a resolution of $1024 \times 512$ into an irradiance map of $512 \times 256$ and a pre-filtered environment map with 6 mipmaps, ranging from $1024 \times 512$ to $16 \times 8$.

### D.2  MATERIAL EDITING

By modeling the avatar's material properties for physical shading, we can easily edit the avatar's materials. In Fig.9, we show material editing under new lighting conditions by gradually increasing

Reconstruct          Material Editing with increasing base fresnel reflectance

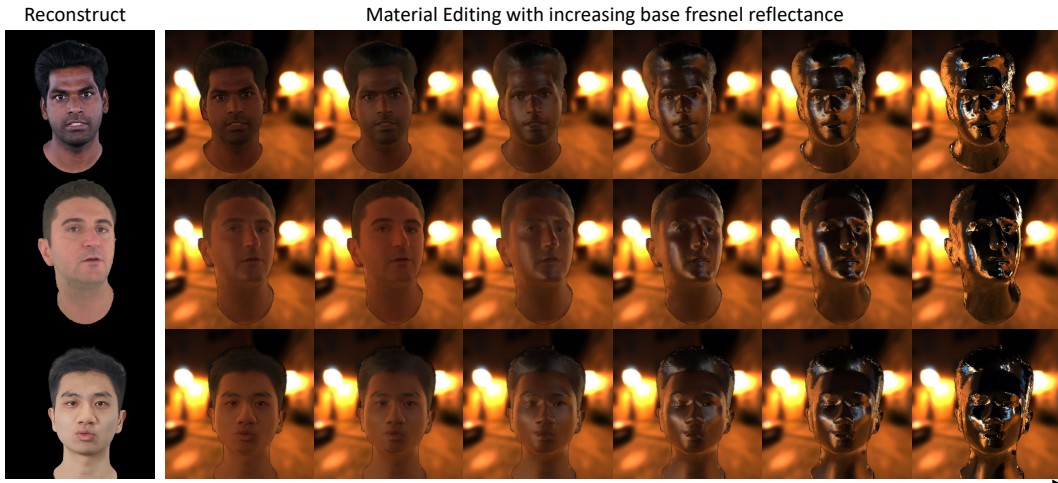

Figure 9: Visual results of material editing. We gradually increase the avatar's base Fresnel reflectance under new environment lighting, enhancing specular reflections. The results align with intuitive expectations, validating the effectiveness of our shading model.

Reference    Reconstruct view                    Novel views

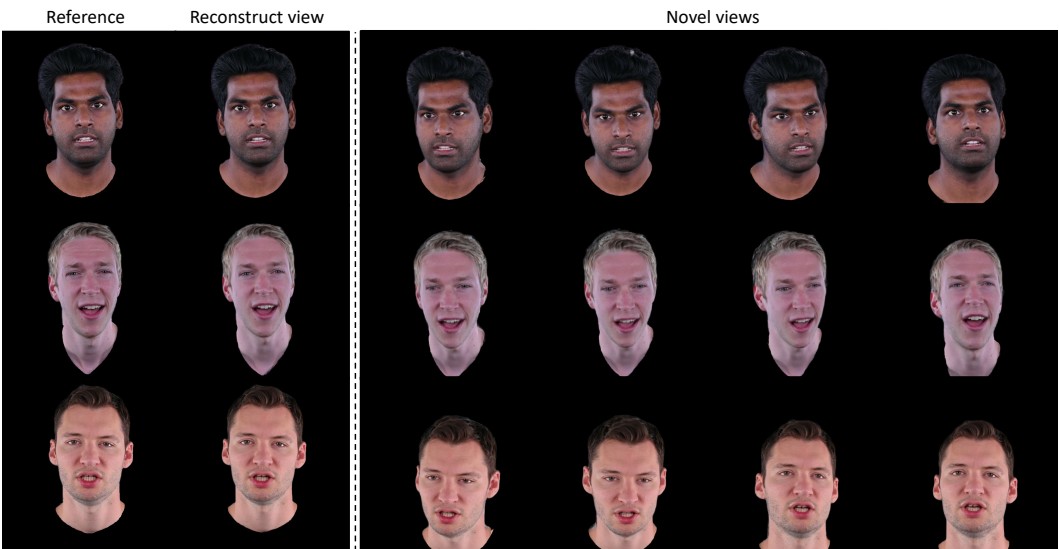

Figure 10: Visual results of novel view synthesis. In each row, the original view of the reconstructed subject is shown on the left, while the rendered novel views are on the right. Our method produces high-fidelity novel views with strong 3D consistency.

the base Fresnel reflectance, which enhances the metallic effect and reduces diffuse reflection. As shown, higher reflectance results in stronger specular reflections, validating the effectiveness of our physically-based shading model.

### D.3 NOVEL VIEWS SYNTHESIS

Although the 3D avatar is reconstructed from a monocular video, it can still render novel views. Fig.10 shows the visual results of our method. As shown, SHARP renders novel views of the head with high 3D consistency and quality, preserving fine texture details.

# E DISCUSSION

## E.1 LIMITATION.

While our method effectively models individual-specific deformations, it remains constrained by FLAME's priors when training data is insufficient. This hinders accurate control of elements like hair or accessories. Moreover, under extreme unseen poses and expressions, performance may degrade, and artifacts may appear in the rendering results. Inaccurate tracking of certain extreme expressions also limits the success of cross-reenactment. Additionally, the use of blendshapes, linear skinning, and shading adds extra computation, slowing down the original 3DGS rendering speed. Offloading these operations to the GPU via CUDA could alleviate this issue. Improvements in these areas offer promising avenues for future research.

## E.2 ETHICAL CONSIDERATIONS.

Creating realistic, controllable head avatars raises concerns about potential violations of portrait rights and privacy. It may also lead to identity theft and misuse in fraud. We strongly condemn any unauthorized use of this technology for illegal purposes. It's crucial to consider ethical implications in all applications of our method to prevent harm to the public.

