# OpenReview forum: "SHARP: Splatting High-fidelity And Relightable Photorealistic 3D Gaussian Head Avatars"
_ICLR.cc/2025/Conference — ICLR 2025 Conference Withdrawn Submission_

### Official Review · Reviewer_xrhj · 2024-10-30

**Soundness:** 2
**Presentation:** 1
**Contribution:** 1
**Rating:** 3
**Confidence:** 5

**Summary:**

Although high-quality head avatars using Gaussian Splatting have emerged recently, this paper addresses the challenges of relighting and tracking inaccuracies. It proposes an end-to-end approach for training a relightable head avatar with material decomposition, incorporating fine-tuning of encoder-based head tracking corrections. This method improves pre-tracked driving signals, such as expression coefficients and jaw pose, which are prone to errors in keypoint-based tracking.

**Strengths:**

1. First Application of Gaussian Splatting for Relightable Monocular RGB Video-based Head Avatars: This work introduces Gaussian Splatting for generating relightable head avatars specifically from monocular RGB video, which has not been addressed in prior methods. This approach allows realistic lighting adjustments from single-camera inputs, setting it apart from previous mesh-based or volumetric avatar models.

2. Use of Off-the-Shelf Albedo Extraction Models: Off-the-shelf albedo extraction models were selected for their straightforward integration and sufficient accuracy in providing base textures for relighting. This choice avoids the complexity and computational demands of custom-built solutions, delivering effective relighting without extensive model-specific tuning.

**Weaknesses:**

1. Novelty & Comparison with Concurrent Works

GaussianAvatars: Given its concurrent acceptance at CVPR 2024, GaussianAvatars should indeed be acknowledged, especially if it's found to be more robust than other Gaussian Splatting-based methods. Including a comparative analysis on performance metrics where possible would be ideal. If GaussianAvatars was excluded due to scope or dataset limitations, clarifying this in the text could help justify its omission. I suggest the authors to augment comparison in terms of rendering quality such as PSNR, LPIPS, SSIM, and L1 in Tab. 1.

FLARE: As this work shares lots of similarities in techniques such as the split-sum approach, Burley's microfacet BRDF, and screen-space deferred shading, a citation and discussion of FLARE is essential. Comparing both contributions and approaches will not only address potential overlap but also reinforce your contributions by contextualizing advancements or modifications. Please clarify the technical difference with FLARE (SIGGRAPH Asia 2023), not just the rendering representation (e.g., Mesh).

Personalized Linear Blend Skinning (LBS): This paper’s deformation strategy, replicating the LBS process of FLAME (3DMM), (in Section 3.3) would benefit from a comparative outline with IMAvatar and PointAvatar. These works have explored incorporating LBS in point-based (even though isotropic points) or other deformable avatar models, notably through backward mapping and occupancy fields. Directly referencing how the deformation principles align or diverge from methods in PointAvatar and MonoGaussianAvatar will emphasize the novelty or refinements of the proposed approach.

2. Verification of Relighting Ability

A quantitative metric to assess relighting quality, such as pixel-wise cosine similarity for normals (as seen in IMAvatar and FLARE), would enhance the robustness of results. Using the FaceTalk dataset or a similar resource could help in aligning with established evaluation standards and provide more depth to the material intrinsic quality analysis.

3. Related Work Positioning

Grassal et al. (2022): Clarify that while Grassal et al. falls under Implicit Neural Representations, it does not rely on a volumetric radiance field. Instead, the approach focuses on vertex-based color prediction with mesh representation, distinguishing it from NeRF-based methods. Repositioning or rephrasing its role in the paper will prevent miscategorization and maintain accuracy.

4. Readability and Formatting Consistency

Ensure uniform spacing between words and citations. Specific lines (such as 139 and 142) should be rechecked for this formatting consistency to improve readability and maintain professionalism in presentation.

**Questions:**

Line 225: Why does only the jaw pose tend to overfit? How does this compare to the expression parameters? Is there any evidence supporting this observation?

Line 290: Why is the reflection direction defined using the view direction and normal reflection? Is there a specific reason for this choice?

---

### Official Review · Reviewer_GsvX · 2024-10-30

**Soundness:** 3
**Presentation:** 3
**Contribution:** 3
**Rating:** 5
**Confidence:** 5

**Summary:**

The input of this paper is a monocular video, and they propose to train an encoder in an end-to-end manner to extract more precise parameters including learnable blend shapes and linear blend skinning. Meanwhile, they employ physically-based shading for appearance modeling. Therefore, their method can do realistic relighting.

**Strengths:**

Actually, the whole pipeline is quite standard. It includes the blendshape representation and linear blend skinning. The strength part is that they make all these parameters learnable. Although making these learnable is also quite standard, they make the whole framework work well and achieve good results.

**Weaknesses:**

Making all these parameters learnable, it brings the good part that the results are better while the bad part the whole optimization takes longer time. It might limit its applications in some scenarios.

**Questions:**

The comparison with some very relevant references are missing, like: Relightable 3D Gaussian: Real-time Point Cloud Relighting with BRDF Decomposition and Ray Tracing.

---

### Official Review · Reviewer_Yd9B · 2024-11-03

**Soundness:** 3
**Presentation:** 2
**Contribution:** 2
**Rating:** 3
**Confidence:** 4

**Summary:**

This paper presents a method to reconstruct high-fidelity, relightable 3D head avatars from monocular video input. The authors claim three main contributions: the integration of learnable blendshapes and linear blend skinning to improve geometric deformation flexibility (but see below regarding concerns about novelty), as well as the application of physically-based shading for realistic relighting under various lighting conditions. The authors conducted experiments on multiple datasets, evaluating the method’s ability to capture detailed textures, facial expressions, and relighting effects across different head poses.

**Strengths:**

The physically based shading based on decomposed head’s appearance effectively contributes to the controlled relighting. This enhances the avatar’s visual quality under different lighting conditions. This is a new feature in head avatar creation.

**Weaknesses:**

-	Technical Novelty. The key technical contributions of this paper are unclear. The main design is very similar to “3D Gaussian Blendshapes for Head Avatar Animation” (SIGGRAPH 2024), except for the integration of relighting components. Also, in the current writing, these similar design elements are all placed in the Methodology Section (Section 3). To improve clarity, the authors should distinctly separate previous designs from their proposed innovations, minimizing potential confusion.
-	Insufficient Comparison. The results appear similar to those of the above SIGGRAPH24 paper.  It also lacks comparisons with recent SOTA models, such as “GaussianAvatars: Photorealistic Head Avatars with Rigged 3D Gaussians (CVPR 2024)” and  “Gaussian Head Avatar:
Ultra High-fidelity Head Avatar via Dynamic Gaussians (CVPR 2024)”.
-	Inconsistent Experimental Results. Maybe the experiment settings are different, but the reported performance metrics from existing works are generally lower than those in their original papers. Please justify this discrepancy.

**Questions:**

Please justify the concerns raised in section of weaknesses

---

### Official Review · Reviewer_Jg7a · 2024-11-05

**Soundness:** 4
**Presentation:** 2
**Contribution:** 3
**Rating:** 8
**Confidence:** 5

**Summary:**

The paper presents SHARP, a 3DGS-based approach for relightable and animatable human head generation from monocular video data.

Unlike previous dynamic 3DGS approaches for human head generation, SHARP learns a blenshape and linear blend skinning rig in an end-to-end optimization framework. This allows for better modeling of the human head and reduces the inherent tracking errors produced by former off-the-shelf 3D face tracking-based systems.

To improve reconstruction under different lighting conditions and allow for relighting applications, this paper models the photorealistic head appearance, even for unconstrained scenarios, by incorporating intrinsic priors, namely albedo, roughness, and Fresnel reflectance, which are combined with a physically-based shading model to approximate the BRDF and account for environment lighting.

SHARP attains remarkable quantitative and qualitative results, outperforming existing baseline methods, especially when reconstructing challenging facial features, such as the mouth interior, hair, and eyes, and transferring expressive facial performances. SHARP also achieves convincing relighting results for different environment maps. As such, the proposed method advances the SoTA in reconstructing high-fidelity animatable and relightable 3D human head avatars.

**Strengths:**

- Proposed approach learns a blendshape and linear blend skinning model within an end-to-end optimization framework for improved tracking consistency and quality, thus obtaining more accurate 3D head reconstruction and reenactment results.
- Use of a physically-based shading model that leverages intrinsic priors, namely albedo, roughness, and Fresnel reflectance to represent complex scene reflectance on a human head.
-  Proposed approach achieves SoTA performance based on extensive qualitative and qualitative evaluations.
- High-fidelity reconstruction of complex facial regions, such as the mouth interior, eyes, and hair.
- Believable relighting effects on a variety of environment maps with mid complexity.

**Weaknesses:**

- The proposed method is a large and complex optimization module with quite some dependencies, e.g., FLAME, SMIRK tracker, 3DGS, Disney’s BRDF model, and Epic’s SplitSum environment map approximation, among others – all having different assumptions. As such, the approach would be non-trivial to implement and replicate, even for experts in the field, without the source code or an in-depth algorithmic description, e.g., in the supplementary material. Providing the source code would be highly appreciated for advancing the field.
- The paper lacks many implementation details, which are needed for reproducibility purposes. For instance, the initialization of deformable human head model parameters, as well as face intrinsic maps, is unclear. It is also unclear if some parameters are regularized or simply converge due to low learning rates, and/or good initializations and (neural) priors. The description of some learnable parameters and/or how they are derived is ambiguous or obscure. Please refer to Questions section for an extensive list of concerns/questions.
- Some design choices, such as optimizable parameters,  properties of materials and deformation basis seem arbitrary, not validated or unknown. See Questions section for specific concerns.
- Limitations should be discussed in the main paper, and a more comprehensive analysis of tracking and relighting scenarios should be included to understand the capabilities or potential limitations. Please refer to Questions section for specific concerns.

**Questions:**

*Section 3.1: Preliminary
- Move this section into the supplementary section as many more researchers know about 3DGS. That would save space for more results and a Limitations section.

*Section 3.2
- Do you finetune encoder E entirely or just the layers that estimate the expression and jaw pose parameters?
- Why are pose and expression parameters not optimized as well? You could start optimizing them after the end-to-end model converges, i.e., after shape, expression, and pose bases are learned, and regularize their values by using the pre-tracked parameters.

*Section 3.3: Geometric deformation modeling
- How are the shape and pose bases initialized? And how do you ensure they represent shape and pose variations? These bases can easily mix up and lose any semantic meaning if not regularized.
- How do you ensure expression parameters are in the expected deformation range? Do you ensure the encoder adapts slowly to improve the estimation of expression parameters?
- Do you initialize Gaussian points via barycentric linear interpolation of FLAME face meshes? Please be more specific.
- For better reproducibility, please elaborate more on the initialization of the blendshape basis and parameters as well as the rotation and scale in the supplementary material.
- Why do you not need to regularize the blendshape basis? I would at least expect an orthogonality constraint between the shape and blendshape basis. Are you setting a slow learning rate to allow slow changes on a blendshape basis?
- Regularizing the estimated albedo map A with the ground truth albedo is not a good idea as it depends on the initial tracking, which might be very poor. Generally speaking, you want to preserve the spatial smoothness properties and spatial consistency of the pseudo ground truth albedo, but not restrict their values to it.

*Section 3.4: appearance modeling
- Please provide more details on how I_irr is generated
- How are the roughness O and reflectance F_0 maps initialized?
- Why are regularizations on irradiance I_irr and reflectance F_0 not needed?
- Where do you get the depth derivates from to derive \hat{N}?

*Section 4.3: Ablation study
- Without PBS: Please mention that the results are provided in Table 2 and include qualitative results in the supplementary material.

*Section 4.4:  Applications
- Please show more relighting results with more complex environment maps in the supplementary material. For instance, show results with bright outdoor lighting having occluding objects casting shadows in part of the scene and indoor lighting with multiple light sources.

*Section E.1: Limitations
- Move this section into the main paper and provide examples of failure cases, such as extreme pose and expression reconstruction, and reenactment and control of hair and accessories.
- Further discuss other potential limitations, such as fast motions, dynamic white and RGB lighting, and strong directional shadows, e.g., half of the face darkened.

*Miscellaneous
- Line 028: Replace “valuable task” with “primal task in photorealistic head digitization for filmmaking, gaming, and telepresence”
- Line 369: Replace “three metrics” with “four metrics”

**Details Of Ethics Concerns:**

Minor concerns about potential societal negative impact and ethical considerations. Please add an extended section about potential misuse scenarios, e.g., Deepfakes. Particularly, discuss additional mitigation strategies or guidelines for responsible use of the technology and detection of synthetically generated results.

---

### Note · Authors · 2024-11-14

I have read and agree with the venue's withdrawal policy on behalf of myself and my co-authors.